# Social rank-order stability of mice revealed by a novel food competition paradigm in combination with available space competition paradigms

**Meiqiu Liu[1], Yue Chen[1], Rongqing Chen[2,3]***

[1]Guangdong Province Key Laboratory of Psychiatric Disorders, Department of Neurobiology, School of Basic Medical Sciences, Southern Medical University, Guangzhou, China; [2]The National Key Clinic Specialty, The Engineering Technology Research Center of Education Ministry of China, Guangdong Provincial Key Laboratory on Brain Function Repair and Regeneration, Department of Neurosurgery, Zhujiang Hospital, Southern Medical University, Guangzhou, China; [3]Key Laboratory of Mental Health of the Ministry of Education, Guangdong-Hong Kong-Macao Greater Bay Area Center for Brain Science and Brain-Inspired Intelligence, Department of Neurobiology, School of Basic Medical Sciences, Southern Medical University, Guangzhou, China

*For correspondence:
creatego@hotmail.com

**Competing interest:** The authors declare that no competing interests exist.

## eLife Assessment

This study proposes a **useful** assay to identify relative social ranks in mice incorporating the competitive drive for two basic resources - food and living space. Using this new protocol, the authors provide **solid** evidence of stable ranking among male and female pairs, while reporting more fluctuant hierarchies among triads of males. The evidence is, however, limited by the lack of ethologically based validation, assessment of the influence of competitor recognition, and proof of concept of application to neuroscience. This manuscript may be of interest to those interested in social behavior and related neuroscience.

**Abstract** Psychological, behavioral, and biological studies on social organization and competition with animal models are boosting. The mouse has been recognized as a valuable and economic model animal for biomedical research in social behaviors; however, currently available food competition paradigms for mice remain limited. Discrepant paradigms involving different competitive factors, such as physical strength vs psychological features, muscular confrontation vs threat perception, and boldness vs timidity, may produce task-specific win-or-lose outcomes and lead to inconsistent ranking results. Here, we developed a food competition apparatus for mice, in which contenders were a pair of mice eager to take over the same food pellet placed under a movable block in the middle of a narrow chamber where they were separated to either the right or left side. This food pellet competition test (FPCT) was designed to (1) provide researchers with a choice of new food competition paradigm and (2) expose psychological factors influencing the establishment and/or expression of social status in mice by avoiding direct physical competition between contenders. Meanwhile, we wanted to evaluate the consistency of social ranking results between FPCT and typically available space competition paradigms—tube test and warm spot test (WST). We hypothesized inconsistency of rankings of mice tested by FPCT, tube test, and WST as they possess different targets for mice to compete and different factors determining competitiveness.

Interestingly, application of FPCT in combination with tube test and WST discovered unexpected consistency of mouse social competitivity and rankings in a grouped male or female mice that were housed in either a two- or three-member cage, most likely indicating that the status sense of animals is part of a comprehensive identity of self-recognition of individuals in an established social colony. Furthermore, the FPCT may facilitate research on social organization and competition, given its reliability, validity, and ease of use.

## Introduction

For gregarious animals and human beings, both competition and cooperation between individuals or populations are fundamental social behaviors important for the survival and evolution of the collective strains and species. Social competitions occur naturally to determine the ownership or priority of living territory, food, water, mates, other resources, and non-resource requirements (*Shimamoto, 2018*; *Korzan and Summers, 2021*). Through these competitions, social hierarchies from the dominant to the subordinate are established within a living group, where individuals at higher social status are granted corresponding priorities of resources and non-resource requirements like living space, food availability, reproduction, and safeguard (*Shimamoto, 2018*; *Korzan and Summers, 2021*). The establishment of dominance hierarchies reduces the intensity and frequency of mutual aggression within groups and strains, so as to maintain their inner stability and fulfill outer assignments (*Guo et al., 2020*). Impaired awareness of social competition has been documented in individuals with autism spectrum disorder (*Gates et al., 2017*; *Su et al., 2022*), and reduced social interaction has been characterized in corresponding animal models (*Sato et al., 2023*). Similarly, maladaptive responses to social status loss have been associated with patient depressive disorders (*Price et al., 1994*; *Komori et al., 2019*) and animal models of depression (*Fan et al., 2023*; *Battivelli et al., 2024*).

Although humans and primates exhibit complex social interactions that are relatively easy to observe, their use for biological research is practically limited. In contrast, rodents—particularly mice and rats—have been emerging to serve as valuable and cost-effective models for studying the biological basis of social cognition, social interaction, and social organization (*Fan et al., 2023*; *Wang et al., 2011*; *Zhou et al., 2017*). Among the social competing objects, resources of living space, water, and food are fundamentally essential for the survival of animate beings. Therefore, living space, water, and food competitions are most frequently used to investigate social competitive behaviors and hierarchical ranking of rats or mice (*Korzan and Summers, 2021*; *Goodman et al., 2021*; *Lucion and Vogel, 1994*). The tube test is a simple and reliable method for assessing social hierarchy by simulating the competition for living space among animals. In the tube test, the dominant mice largely consistently squeeze out the weaker ones from the narrow tube, indicative of an overwhelming space demand of mice at higher rank (*Fan et al., 2019*). Warm spot test (WST) is another space competition paradigm where a pair of mice compete for a pitifully small warm corner in a freezing cage (*Zhou et al., 2017*; *Zhu and Hu, 2018*). Some food competition tests have also been applied to animals such as rats (*Cordero and Sandi, 2007*; *Jupp et al., 2016*; *Timmer and Sandi, 2010*), chickens (*Lee et al., 1982*), pigs (*Hessing and Tielen, 1994*), and mice (*Yoon et al., 2022*; *Shin et al., 2022*; *Merlot et al., 2004*; *Padilla-Coreano et al., 2022*). Larger body size is an advantage of rats, chickens, and pigs that makes detecting behavioral patterns easier by the experimenters or monitoring video, but the food competition procedures for those animals are difficult to be applied to mice due to their much smaller size. Both mice and rats are rodents and mostly used experimental animals, but rats are more socially tolerant and less hierarchical than mice (*Kim et al., 2015*; *Bartos and Brain, 1994*).

Relative to space competition, food competition tests for mice have been designated and applied less commonly in animal studies despite its long history (*Fredericson, 1950*; *Fredericson, 1952*; *Schuhr, 1987*). Several issues could be thought to be the underlying limitations for the application of food competition paradigms. First, there are methodological issues in some of these approaches, such as long video recording duration and difficulty in analyzing animals' behaviors during competitive physical interaction in videos, hindering their application by laboratories that cannot afford sophisticated equipment (*Shin et al., 2022*; *Merlot et al., 2004*; *Li et al., 2007*; *Timmer and Sandi, 2010*; *Löfgren et al., 2013*; *Costa et al., 2021*) and analysis (*Li et al., 2022*). Second, aggressive behaviors often occur during competition in these approaches as the mice compete in a shared arena. Third, the prolonged food deprivation is usually required to increase the effortful food competition. For the

second and third issues, on one hand, the aggressiveness/being aggressed and intense food deprivation cause stress responses and changes of physiological state of animals (*Battivelli et al., 2024*; *Merlot et al., 2004*; *Fredericson, 1950*; *Fredericson, 1952*). On the other hand, competition in the same arena conflates multiple determinants contributing to the outcomes of competition involving physical aspects—such as muscular strength, vigorousness of fighting, bite wounding—and psychological aspects—such as boldness, focused motivation, active self-awareness of status. This feature makes interpretation of the outcomes of competition difficult to discriminate the dominant behaviors and aggressive behaviors. So far, the food competition tasks had not been conceived to separate the physical aspects from psychological aspects interpreting the mice's winning/losing.

Therefore, to have a new choice of food competition paradigm for mice, and to facilitate the exposure of psychological aspects contributing to the winning/losing outcomes in competitions, we generated a convenient, easily operative, and peaceful food competition paradigms for a pair of mice competing for a food pellet after mild calorie restriction. The food pellet competition test (FPCT) was purposed for mice food competition without physical contact during competition, but tube test and WST were of space competition during which the mice need direct physical contact. Thus, we expected inconsistent evaluation results of competitiveness and rankings if we compared FPCT with typically available competition paradigms—tube test and WST.

## Results

### FPCT did not detect significant difference in the winning/losing results between unfamiliar non-cagemate male mice

The experimental procedure of FPCT consists of habituation, training, and test (*Figure 1A*). After handling the mice in the homecage and experimental room, the mice were trained to enter the arena alternately from the left and right gates to be familiar with the chamber arena (*Figure 1B*, *Figure 1—figure supplements 1 and 2*), followed by the next step to find a small food pellet in the trough in the middle of the chamber floor (*Figure 1C*, *Figure 1—figure supplement 2*). At the end of this step of training, the mice would go directly for the food pellet when they entered the arena (*Figure 1—video 1*). Then, a movable block was hung up at the roof of the chamber (*Figure 1D*, *Figure 1—figure supplements 1 and 2*). The bottom of the high rectangular block was positioned nearly right above the trough so that the mice could not possess the food pellet unless they pushed the block away. The block is transparent and dug with holes at the lower part, enabling the mice to see and smell the food pellet (*Figure 1D*). The mice were trained to learn to move away the block and reach the food pellet twice daily (*Figure 1D*). Usually, it took 4–5 days for the mice to learn to walk straightforwardly to the obstacle once they entered the chamber and move the block easily to expose the food pellet (*Figure 1E*, *Figure 1—video 2*). The orientation of the entries did not generate bias of performance of mice in the apparatus (*Figure 1E*).

In the test section, two mice were allowed to enter the separated rooms of the chamber simultaneously from one of the two entries, competing for the food pellet either placed in the trough below the block (test 1, *Figure 1F*)—where the pellet was accessible if the block was pushed away—or on the inner bottom of the block (test 2, *Figure 1G*)—where the pellet was inaccessible. The two mice were able to see and possibly sniff each other through the transparency and holes of the block, as well as the gap between the block and chamber.

In the first experiment utilizing the FPCT, we tested age- and weight-controlled unfamiliar non-cagemate male mice to examine whether there was obvious nonsocial priority-associated competitiveness expressed in the FPCT. Eight single-housed mice were trained and then randomly divided into groups A and B for competition test 1. Within the 4 consecutive days of competition test, there were a total of 16 trials, as one mouse in a group matched one of the mice from the other group once daily and the matched two players participated in the contest only once in the 4-day schedule (*Figure 1H*). In each trial of test 1, the mouse that obtained the food pellet was credited a score of 1 (time of winning), while the mouse that failed to obtain the food pellet was credited a score of 0 (*Figure 1I*). Statistical data found that daily average scores acquired by the two groups of mice were not significantly different (*Figure 1J*), illustrating that no nonsocial priority-associated competition factor was obviously involved in the FPCT with the sex-, age-, and weight-controlled unfamiliar mice.

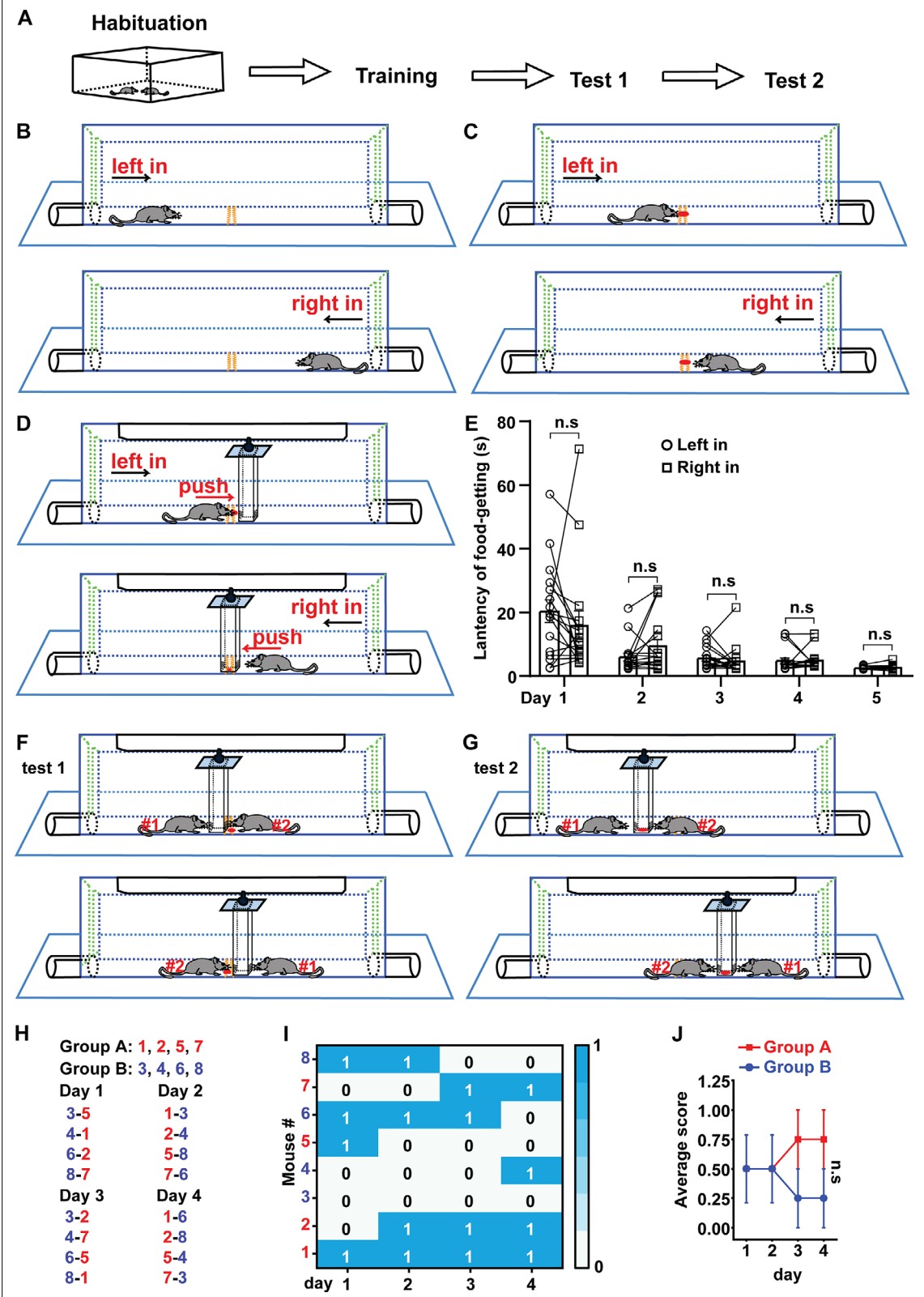

**Figure 1.** Schematic drawing of food pellet competition test (FPCT) procedure and measuring the non-cagemate male mice based on the winning/losing outcomes of FPCT. (**A**) The overall procedure of FPCT experiment. (**B**) Habituation to the arena where the mice entered alternately from left and right sides. (**C**) Training to find food pellet without the existence of the movable block. (**D**) Training to get the food pellet placed under the movable block. (**E**) Statistical data of latency of food-getting (the time from entering the arena to eat the food) showing the progress of training in (**D**). Paired

*Figure 1 continued on next page*

*Figure 1 continued*

Student's t-test, n=17, n.s. stands for nonsignificant difference. (**F**) Direct win-lose test via analyzing food occupation (test 1). (**G**) Indirect win-lose test via analyzing mouse's attempt to gain the food (test 2). (**H**) Eight single-housed mice were numbered and randomly divided into groups A and B after being trained. One of the mice in group A matched only once with one of the mice in group B in the 4 days of test 1. The number listed on the left-right indicated the corresponding mice would enter the arena from left-right entry. Scores 1 and 0 indicated winning and losing the food competition, respectively. (**I**) Heatmap showing the competition outcomes of the non-cagemate male mice ranked with FPCT. (**J**) The scores acquired by each mouse in the competition were averaged in each day for the two groups. Unpaired Student's t-test = 4, n.s. stands for nonsignificant difference. Data were represented as mean ± SEM.

The online version of this article includes the following video, source data, and figure supplement(s) for figure 1:

**Source data 1.** Statistical behavioral data.

**Figure supplement 1.** Photographs of food pellet competition test (FPCT) setup.

**Figure supplement 2.** Schematic illustration of the assembling of the food pellet competition test (FPCT) setup.

**Figure supplement 3.** Schematic drawing to show a tip to reduce the friction between the pulley and the track.

**Figure supplement 4.** Schematic drawing to show the marking of the starting state when the mouse's four legs just entered the chamber arena.

**Figure 1—video 1.** In step 3 of training, the mouse was trained to know the position of the food pellet in the absence of the movable block.
https://elifesciences.org/articles/103748/figures#fig1video1

**Figure 1—video 2.** In step 4 of training, the mouse was learning to get the food pellet placed in the trough of the chamber floor nearly under the movable block.
https://elifesciences.org/articles/103748/figures#fig1video2

**Figure 1—video 3.** In the food pellet competition test (FPCT) test 1, the mice were competing for the food pellet.
https://elifesciences.org/articles/103748/figures#fig1video3

**Figure 1—video 4.** In the food pellet competition test (FPCT) test 2, the mice were attempting to get the food pellet placed in the inner bottom of the block.
https://elifesciences.org/articles/103748/figures#fig1video4

## FPCT identified different social ranks between two-cagemate male mice, verified by tube test

Social hierarchy or status of many animals is largely arranged by the outcomes of winning/losing social competitions, manifesting as dominant or subordinate behaviors in a well-established social colony. To see the efficacy of FPCT in detecting the social status of mice in a well-established social colony, we conducted 4 days of FPCT test 1 containing one trial per day for each of the 16 pairs of cagemate male mice after training (*Figure 2A*). In the neighboring 2 days, each mouse was allowed to enter the chamber from different entries. In each trial, the mice were scored 1 for winning and 0 for losing the food pellet. In the final trial, the mouse obtaining the food pellet was ranked #1 and considered the final winner, while the mouse failing to get the food pellet in the match was ranked #2 and claimed as the final loser. Retrograde analysis of the time spent to get the food after entering the arena gate in the last day of training showed that the winners and losers displayed the same level of training to get the food, indicating equivalent motivational craving for the food (*Figure 2A*, day –1). However, when there was competition in the test, the winners tended to win the pellet over all the four trials (*Figure 2A*, days 1–4). Notably, the presence of an opponent during the test did not significantly change the latency to get the food, comparing with the last day of training when there was no opponent (*Figure 2A*). In all the 16 pairs of mice, only 1 pair exhibited alternating winning/losing outcomes with an inter-trial consistency at 50% (*Figure 2B*), while all the other 15 pairs kept 100% consistency of winning/losing readouts (*Figure 2C*). Averagely, the competition results were stable throughout the 16 trials (*Figure 2D*), and a rate of consistency reached 96.88% (*Figure 2E*). In test 2 (indirect win-lose test via analyzing mouse's attempt to gain the food), we observed that the winners determined by the final trial of test 1 tried harder to get the food pellet by spending more time to push the block within the single 2 min test, a limited time to avoid distinction of attempt (*Figure 2F*). Thus, these data demonstrate that FPCTs are effective to rank the established social status of mice, and that consecutive four trials (one trial daily) might be least required for the FPCT to achieve readout consistency and reliability.

Many factors may contribute to the competitive outcomes in mice, such as age, sex, training level, physical strength, and intensity of psychological factors. We were interested to understand which

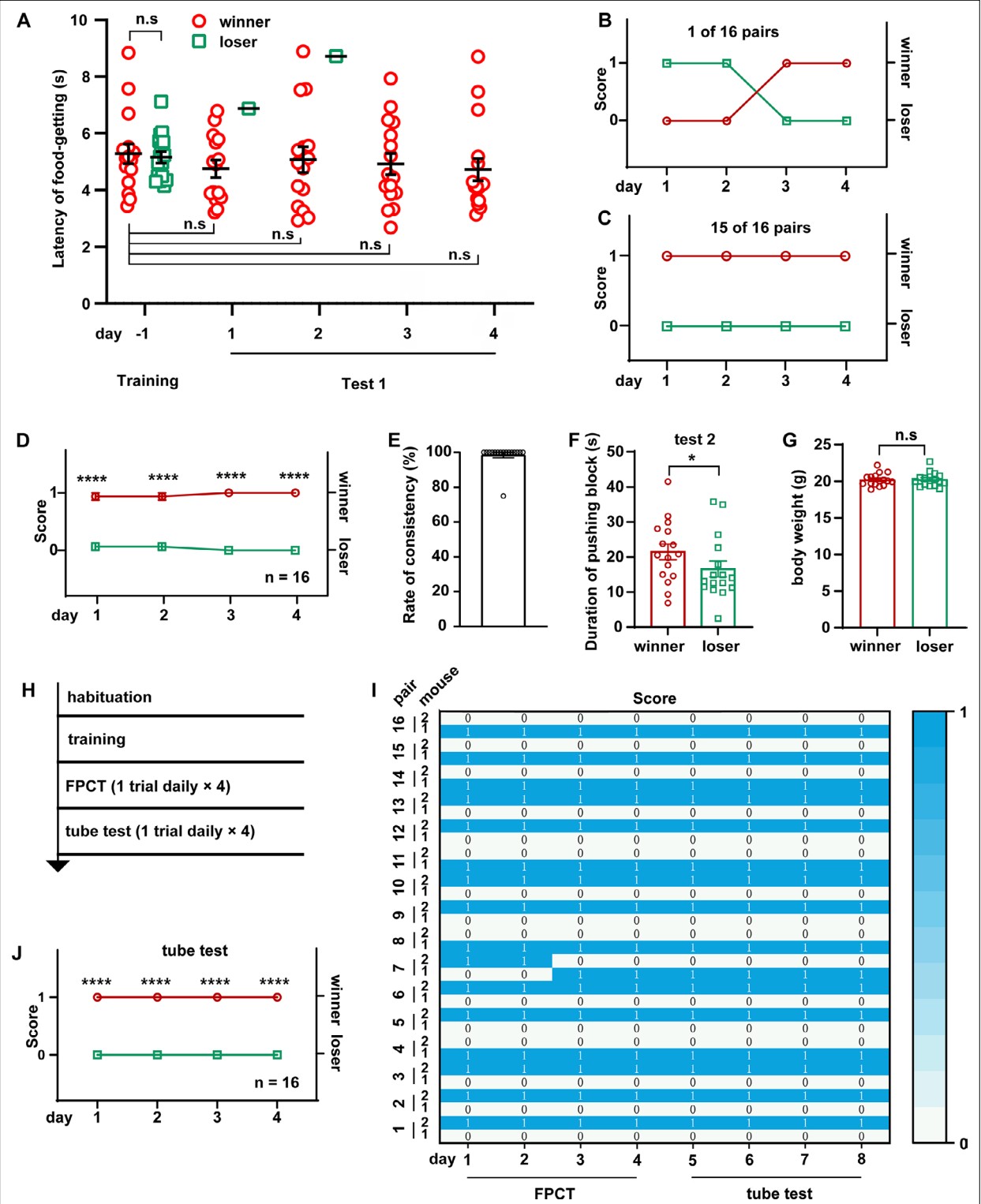

**Figure 2.** Ranking the two-cagemate male mice with food pellet competition test (FPCT) and tube test. (**A**) Statistical data showing the latency of food-getting from the last day of training until the end of test 1. Two-way ANOVA multiple comparisons, n=16 pairs, n.s. stands for nonsignificant difference. (**B**) 1 of the 16 pairs of cagemate male mice exhibited alternating winning/losing outcome in FPCT test 1. (**C**) 15 of the 16 pairs of male mice exhibited fully congruent winning/losing outcome in FPCT test 1. (**D**) Statistics of ranking of male mice over four trials (one trial daily) of FPCT test 1. Two-way ANOVA, n=16, ****p<0.0001. (**E**) Average rate of consistency within trials. For each pair, the rate of consistency was calculated as the percentage of the number of trials (in all four trials) resulting in the outcome same as the fourth trial (n=16). (**F**) Statistics of duration spent on pushing block in FPCT test

*Figure 2 continued on next page*

*Figure 2 continued*

2. The winner or loser identity was determined by the last trial of test 1. Paired Student's t-test, *p<0.05, n=16. (**G**) Body weight of male mice measured after FPCT. Unpaired Student's t-test, n.s. stands for nonsignificant difference, n=16. (**H**) Timeline of experiments showing tube test was conducted after FPCT. (**I**) Heatmap showing the outcomes of social competition of paired male mice ranked with FPCT and tube test. (**J**) Statistics of ranking of male mice over four trials (one trial daily) of tube test. Two-way ANOVA, n=16, ****p<0.0001. All statistical data were represented as mean ± SEM.

The online version of this article includes the following source data for figure 2:

**Source data 1.** Statistical behavioral data.

factor determines the outcomes of mice's winning/losing in the FPCT match. First, effects of age and sex on winning/losing outcomes were excluded as the mice were age-matched and solely males. Second, the weight of mice was also excluded to determine the competency possibly by providing physical strength of pushing, as the weight difference between paired contender mice before training was controlled in less than 10%, and no difference between the winners and losers in weight measured again after finishing the food competition tests (*Figure 2G*). Third, biased training level should produce distinct likelihood of winning and losing, but retrospective comparison of the duration they took to obtain the food pellets in the last day of training—when there was no competition—reported no difference in the latency to get the food pellet between the winners and losers (*Figure 2A*), indicating that the mice were equally trained and motivated. Fourth, the major physical contribution to the competition based on violence, aggressiveness, and fighting strength was minimized in this competitive scenario due to the separation of the mice by the obstacle. Thus, we postulated that the winning/losing outcomes in this paradigm mainly rely on mice's state of psychological factors, e.g., intensity of motivation, self-awareness of social status, memory of winning/losing experience in the homecage, fear of revenge when returning the living cage. Most likely, self-awareness of social status plays a determinant role in the outcomes.

To see whether social ranking revealed by the FPCT is consistent with the tube test that represents the simple and robust behavioral assay for space competition (*Fan et al., 2019*), we continued the experiments to rank the mice using the tube test following FPCT (*Figure 2H and I*). We found that consistency of outcomes between the FPCT and tube test was 100%, as the winners and losers in the last trial of the FPCT continued consecutively to be the winners and losers, respectively, during the four consecutive trials of tube tests (*Figure 2I and J*). Therefore, the rank order examined by the FPCT is fully verified by the tube test.

## Social ranking of two-cagemate female mice using FPCT, tube test, and WST

It is widely accepted that males, including human beings and animals, are evolutionarily more eager to be dominant, more aggressive, and more hierarchical, but it is arguable regarding whether females have less competition and looser social organization (*Ivan et al., 2023*; *Stockley and Campbell, 2013*; *Williamson et al., 2019*). On the basis of the investigation of aggressive behaviors, it has been reported that the social colony of female mice is also hierarchical (*Schuhr, 1987*; *Smith-Osborne et al., 2023*). Taking use of this newly designed mouse competition task in which dependence of competency on physical aggressiveness is minimized, we examined whether intra-female social status between two cagemate female mice was observable in the FPCT context. Notably, the FPCT test 1 showed that 3 in 10 pairs of cagemate female mice exhibited alternating winning/losing outcomes, while the majority of cagemates (7 in 10 pairs) showed fully congruent grades (*Figure 3A–E*). On average, 10 female pairs displayed stable rankings throughout all trials (*Figure 3F*). The overall rate of inter-trial consistency was 90% (*Figure 3G*), which was not significantly different from that of male mice (p>0.05 comparing data in *Figure 2E and G* using Mann-Whitney U test). In test 2, we observed that winner mice in test 1 made more efforts trying to get the food pellet (*Figure 3H*). The body weights examined after the tests (*Figure 3I*) were similar between the winners and losers, so the weight parameters were not likely to contribute to the competency. Interestingly, retrospective analysis of training data displayed similar training level of food-getting and craving state for food (*Figure 3A*). However, the presence of an opponent during test 1 could significantly disturb the female mice and increase their latency to get the food, compared with the last day of training when there was no opponent (*Figure 3A*). These FPCT results suggest that the social relationship of female mice is also stratified.

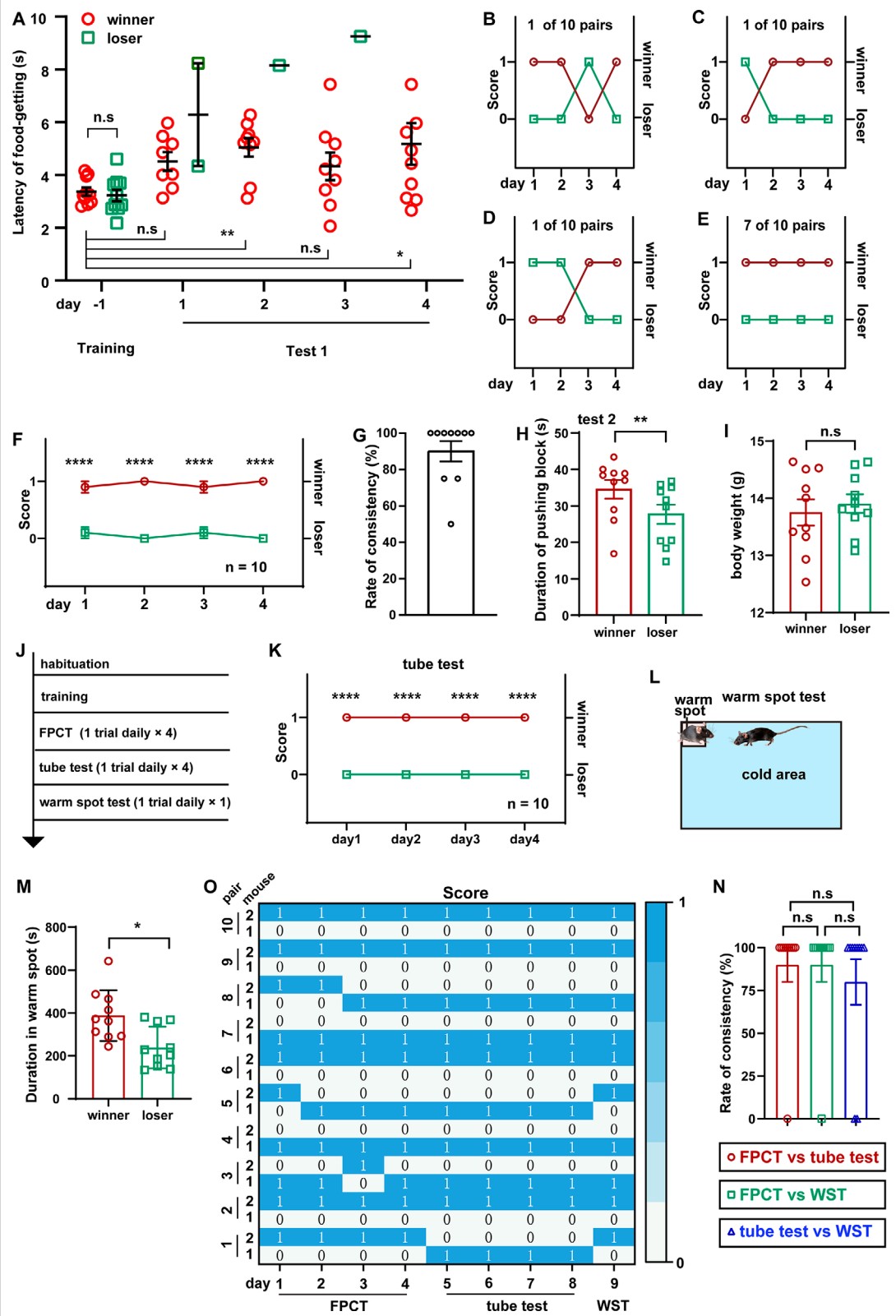

**Figure 3.** Ranking the two-cagemate female mice using food pellet competition test (FPCT), tube test, and warm spot test (WST). (**A**) Statistical data showing the latency of food-getting from the last day of training until the end of the test 1. Two-way ANOVA multiple comparisons, n=10 pairs; *p<0.05, **p<0.01, n.s. stands for nonsignificant difference. (**B–D**) 3 of the 10 pairs of cagemate female mice exhibited alternating winning/losing outcome in FPCT test 1. (**E**) 7 of the 10 pairs of female mice exhibited fully congruent winning/losing outcome in FPCT test 1. (**F**) Statistics of ranking of female

*Figure 3 continued on next page*

*Figure 3 continued*

mice over four trials (one trial daily) of FPCT test 1. Two-way ANOVA, n=10, ****p<0.0001. (**G**) Average rate of consistency within trials. For each pair, the rate of consistency was calculated as the percentage of the number of trials (in all four trials) resulting in the outcome same as the fourth trial (n=10). (**H**) Statistics of duration spent on pushing the block in FPCT test 2. The winner or loser identity was determined by the last trial of test 1. Paired Student's t-test, *p<0.05, n=10. (**I**) Body weight of female mice measured after FPCT. Unpaired Student's t-test, n.s. stands for nonsignificant difference, n=10. (**J**) Timeline of experiments showing tube test and WST was conducted after FPCT. (**K**) Statistics of ranking of female mice over four trials (one trial daily) of tube test. Two-way ANOVA, n=10, ****p<0.0001. (**L**) Schematic of the WST. (**M**) Cumulative duration of mice occupying the warm spot. The winner or loser identity was determined by the last trial of FPCT. Paired Student's t test, n=10, *p<0.05. (**O**) Heatmap showing the outcomes of social competition of paired female mice ranked with FPCT, tube test, and WST. (**P**) Rate of consistency between FPCT and tube test (day 4 vs day 5), FPCT and WST (day 4 vs day 9), as well as tube test and WST (day 8 vs day 9). One-way ANOVA test, n=10, n.s. stands for nonsignificant difference. All statistical data were represented as mean ± SEM.

The online version of this article includes the following source data for figure 3:

**Source data 1.** Statistical behavioral data.

In addition to the tube test, WST, the regional urine marker, and the courtship ultrasound vocalization test are established behavioral methods for determining social rank in mice (*Zhou et al., 2017*; *Lucion and Vogel, 1994*; *Zhu and Hu, 2018*; *Cordero and Sandi, 2007*; *Timmer and Sandi, 2010*; *Arrant et al., 2016*; *Ujita et al., 2018*; *Desjardins et al., 1973*; *Long, 1972*; *Kalueff et al., 2006*; *Dizinno et al., 1978*). Tested animals in different tasks may pursue specific goals and utilize their specific expertise, strength, and skills. Thus, we assumed that different competition paradigms may produce inconsistent social ranking readouts of the same social colony, resulting from various types and levels of competition. To check this assumption, we continued to assess these female mice using tube tests and WST following FPCT (*Figure 3J*). As expected, some of the trials exhibited inconsistent rankings in the three tests. However, most of the trials unexpectedly showed that mice maintained the winner or loser identity acquired in the FPCT in subsequent tube test and WST (*Figure 3K–M*). Overall, no difference was observed in the rate of inter-trial consistency comparing FPCT with tube test, FPCT with WST, and tube test with WST (*Figure 3N*). These data illustrate that mouse social competency and status of established female mice colony are overall stable so that a well-competitive subject exhibits dominance not just in a specific, but in a variety of contexts, despite that the different contexts contain distinct competitive factors like food and space, or competitive aspects like physical strength and psychological factors.

## Social ranking of triads of male mice using FPCT, tube test, and WST

Social organization of the bigger crowd could be much more complicated than the simplest two-member group. It is controversial whether the dominant and subordinate roles in a larger group of mice tend to be mobile or rigorous in a variety of contexts (*Costa et al., 2021*; *Varholick et al., 2019*; *Williamson et al., 2016*). To probe this issue, we raised three male mice in a cage, and then, following adequate habituation and training, we ranked their hierarchies using FPCT, tube test, and WST sequentially (*Figure 4A*). In the FPCT and tube test, the three-cagemate mice contested in a round-robin style as both were designated for two contenders in a time. The ranking result showed that six in nine groups of mice displayed some extent of flipped ranking (*Figure 4B–G*), and only three in nine groups displayed continuously unaltered ranking (*Figure 4H*). On average, in the total of 27 trials consisting of 12 trials of FPCT, 12 trials of tube test, and 3 trials of WST, an obvious stable linear intragroup hierarchy was observed across all the trials and tasks (*Figure 4I and J*). Comparison of inter-task consistency revealed that the ranks assessed by FPCT, tube test, and WST did not differ from each other (FPCT vs tube test, 85.19+17.95%; FPCT vs WST, 81.59+17.95%; tube test vs WST, 81.48+17.95%; p>0.05, one-way ANOVA test; *Figure 4K*). These results, together with the results of female mice tested by FPCT, tube test, and WST, illustrate that the dominant and subordinate roles in an established group of mice are rigorously linearly stratified so that the status can be manifested in various contexts.

## Discussion

Social hierarchy is an innate feature of gregarious animals, usually established through social competition. Among the diverse contents for social competition, resources of living space and food are

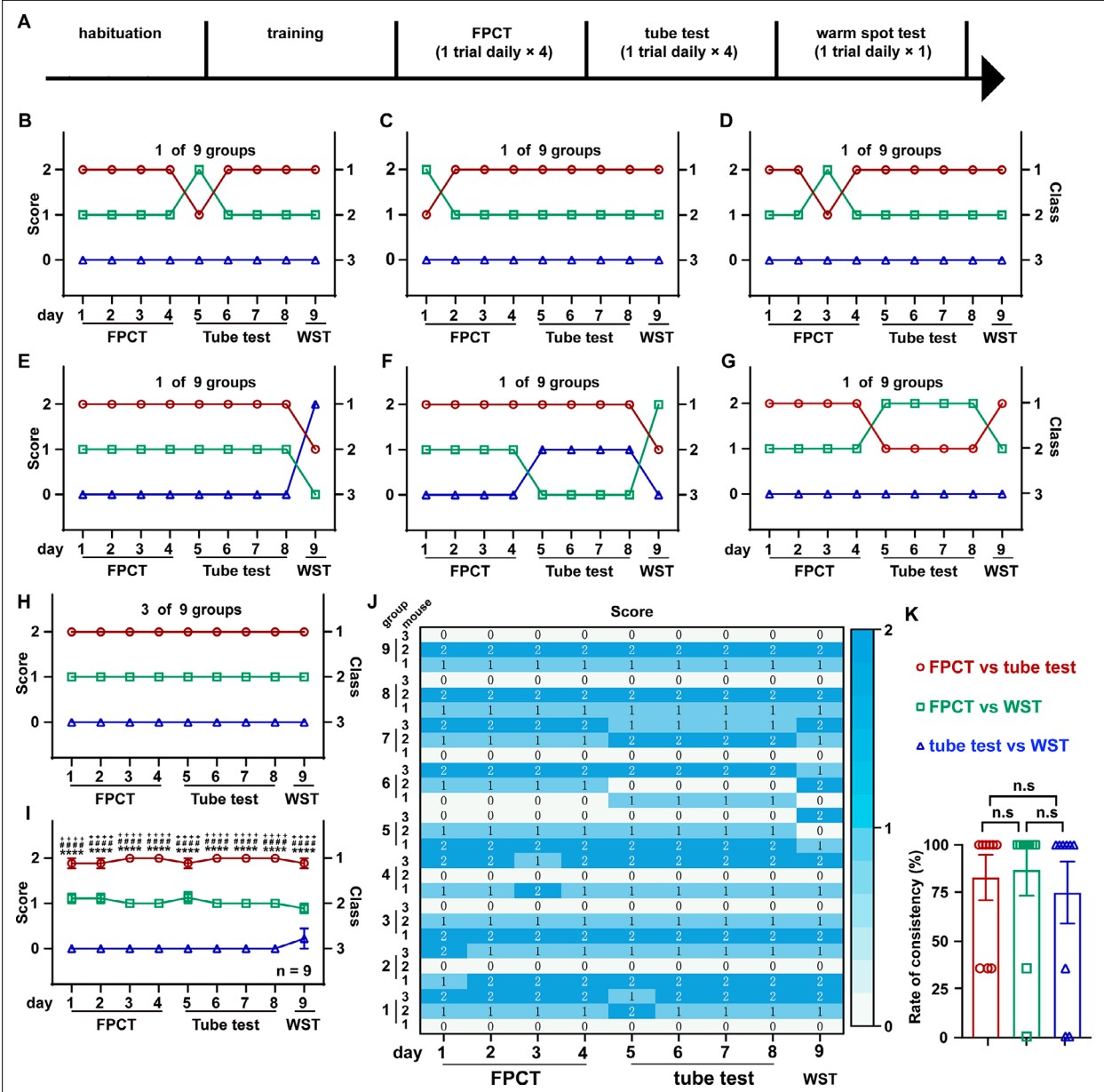

**Figure 4.** Ranking the triads of male mice using food pellet competition test (FPCT), tube test, and warm spot test (WST). (**A**) Timeline of experiments to rank three-cagemate male mice using FPCT, tube test, and WST sequentially. In the FPCT and tube test, the mice contested in a round-robin style within each three-cagemate group. (**B–G**) 7 of the 10 groups of male mice exhibited alternating winning/losing outcomes during the whole competition tasks. (**H**) 3 of the 10 groups of male mice exhibited fully congruent winning/losing outcomes during the whole competition tasks. (**I**) Statistics of ranking of mice over the whole competition tasks. Two-way ANOVA test, n=10, $^{++++}p<0.0001$ comparing FPCT and tube test, $^{****}p<0.0001$ comparing FPCT and tube test, $^{####}p<0.0001$ comparing tube test and WST. (**J**) Rate of consistency between FPCT and tube test (day 4 vs day 5), FPCT and WST (day 4 vs day 9), as well as tube test and WST (day 8 vs day 9). One-way ANOVA test, n=10, n.s. stands for nonsignificant difference. (**K**) Heatmap showing the ranking outcomes of grouped male mice during the whole competition tasks. All statistical data were represented as mean ± SEM.

The online version of this article includes the following source data for figure 4:

**Source data 1.** Statistical behavioral data.

fundamentally critical priority for animals (**Korzan and Summers, 2021**; **Goodman et al., 2021**). Mice have been designated to participate in food competition tests for a long time, since at least the 1950s (**Fredericson, 1950**; **Fredericson, 1952**). In these applied food competition tests, social ranks were scored by calculating the total amount of food consumed by each mouse competing in the same chamber or cage and/or analyzing their physical aggressiveness (**Yoon et al., 2022**; **Shin et al.,**

*2022*; *Schuhr, 1987*; *Williamson et al., 2016*). Some common limitations for the wide distribution of these food competition tests are significant, including the occurrence of aggressive behaviors, requirement of prolonged food deprivation, calculation formulas, long video recording duration, and difficulty in discriminating individual animal behaviors during interacting with group members in videos (*Shin et al., 2022*; *Merlot et al., 2004*; *Li et al., 2007*; *Timmer and Sandi, 2010*; *Weger et al., 2018*; *Löfgren et al., 2013*; *Costa et al., 2021*). Food shortage serves as a strong driver to induce aggressiveness and competitive behaviors (*Tucci et al., 2006*). However, in rodents, being physically aggressed and undergoing intense food deprivation can influence social competitive behaviors via disturbing the internal state (*Merlot et al., 2004*; *Li et al., 2007*; *Márquez et al., 2013*) and triggering robust stress responses (*Tucci et al., 2006*; *Márquez et al., 2013*; *Kennedy and Shapiro, 2009*; *Reppucci et al., 2020*; *Wang et al., 2021*).

Here, we developed and validated a food competition assay—FPCT, a simple and effective tool designed to provide a means of assessing social status of mice, favoring the growing scientific interest in understanding the neurobiological insights of social hierarchy. The potential influence of food restriction on competitive behavior was minimized in our task where the mice underwent only a 24 hr food deprivation period at the beginning of training, followed by restricted food supply to meet basic energy requirements. The aggressive situations, at least direct physical fighting, were prevented during this ranking test due to the separation of the mice on either side of the arena. The FPCT paradigm is conveniently and generally applicable for its simple procedure. First, the mice were housed together to habituate each other and establish hierarchical social structure. Then, the mice were trained individually to accommodate the chamber arena, know the existence and position of the food pellet, and learn to push away the block above the food pellet to get the food. Importantly to note, stable social hierarchy relies on adequate habituation, and reliable ranking test is based on well training. Finally, the mice underwent the ranking test in the chamber, where they were allowed to enter from either the right or the left side of the chamber separated by the movable block to compete for the same food pellet under the block.

Concerning the interpretation of the ranking results, both static vs dynamic and linear vs despotic hierarchy organizations of grouped mice have been documented (*Costa et al., 2021*; *Varholick et al., 2019*; *Williamson et al., 2016*). The diversity of ranking results might be attributed to various factors. For example, different paradigms containing different tasks that may enable different mice to express their proficient competitiveness such as physical advantage, psychological strength, and learning ability. Bodily conflict between contender mice in a shared space could make muscular strength competitive. It is not sure if body size/weight affects social hierarchy as both correlation (*Costa et al., 2021*) and noncorrelation (*Battivelli et al., 2024*; *Williamson et al., 2016*) between weight and hierarchy have been reported. A strategic task possibly promotes a smarter mouse to be a winner in the test even if it may be in a subordinate role at homecage. Besides, group size of experimental animals (*Van Loo et al., 2001*; *Jirkof et al., 2020*), duration of habituation of group members (*Jirkof et al., 2020*), training level, and test protocols (e.g. short-term vs long-term test; *Varholick et al., 2019*) should be considered as common points giving rise to instability of the formation or expression of social hierarchy. In this novel food competition paradigm, we tested pairs and triads of same-sex mice with similar age and body weight, housed in the same cage for at least 3 days. We did not find a significant difference in food priority within pairs of non-cagemates, whereas we detected overall stable winning/losing outcomes of cagemate pairs and triads. Considering that direct bodily competition between mice was avoided in the FPCT by separating the matching pair in either side of the chamber, the stable ranking results of the matches should be mainly determined by the psychological aspects of the contenders. Most likely, it is a reflection of the mouse's self-awareness of social status.

Combining FPCT with typically available paradigms—tube test and WST—we addressed three questions: whether female mice are stably socially structured, whether FPCT is effective to discover hierarchy rankings across multiple cagemates, and whether different social tasks result in discrepant hierarchy rankings. First, we found that the majority of female pairs displayed overall stable hierarchy ranks, consistent with previous findings (*Smith-Osborne et al., 2023*; *LeClair et al., 2021*). We also detected 3 in 10 female cagemate pairs exhibiting fluctuant rankings during the 4-day FPCT test, but it was not significant from 1 in 16 male pairs. It has been revealed by agonistic behaviors in previous work that female social hierarchies are not as steep and despotic as male hierarchies (*Williamson et al., 2019*). The discrepancy between this work and ours might be attributed to differences in mouse

strains, group sizes, housing models, and detecting methods. Second, by means of round-robin tests of multiple cagemates, both FPCT and tube test detected averagely linear intragroup hierarchies, consistent with each other. Third, considering that diverse competitive factors lie in different tasks, the ranking readouts of male pairs and triads examined by FPCT, tube test, and WST were highly consistent with each other, revealing social rank-order stability of mice across the three types of examining tasks. Notably, the fluctuation of the rankings occurs occasionally, suggesting that the social status of mice could be dynamic and transitive on some occasions (*Varholick et al., 2019*). It would be interesting to investigate whether the social hierarchies in larger sizes of mice groups, especially female mice, are more dynamic and transitive. Alternately, the occasional intra-trial and inter-task ranking fluctuations might be a result of incompetency of paradigms to reflect the real intra-colony structure of mice.

In conclusion, our data suggest that hierarchical sense of animals might be part of a comprehensive identity of self-recognition of individuals within an established social colony. Hopefully, the FPCT will facilitate future studies to reveal more detailed properties of social organization, social competition, and their underlying neurobiological mechanisms.

## Materials and methods

### Animals

All animal experiments were conducted according to the Regulations for the Administration of Affairs Concerning Experimental Animals (China) and were approved by the Southern Medical University Animal Ethics Committee. The mice used in this study were 6- to 8-week-old C57BL/6JNifdc purchased from SPF (Beijing) Biotechnology Company (Beijing, China) and were raised in the environment with the relative humidity 50–75%, temperature 22–24°C, and 12–12 hr light-dark cycle. The mice were allowed to freely access food and water unless undergoing food restriction requirements during experiments.

### FPCT setup

The experimental setup consists of a food competition arena (*Figure 1—figure supplement 1*) and a camera. The arena includes a base, a chamber, two entries, two doors, a movable block, a pulley system, a food trough, and the roof (*Figure 1—figure supplement 2*) made up of a batch of wheels and a wheel track for the pulley system, as well as 15 pieces of acrylic plates for other parts listed with dimensions in *Figure 1—figure supplement 2F*. The movable block hung up under the track section separates the arena into the left and right compartments. It is necessary to apply some paraffin oil to the track to reduce the friction between the pulley and the track (*Figure 1—figure supplement 3*). The food trough is positioned in the middle of the chamber floor and nearly under the movable block unless the block is pushed away by mice. Food pellets are tiny delicious milk crackers. Mouse behaviors are monitored by a side view camera with PotPlayer software.

### FPCT procedure

The experimental procedure consists of habituation, training, and test (*Figure 1A*). The details of the procedure are described below.

#### Habituation

Contender pair of C57BL/6JNifdc mice with similar age and weight (the weight difference was less than 10%) were housed in the same cage for at least 3 days under a 12–12 hr light-dark cycle. Before the formal training of mice, each mouse was handled by experimenters for 3–5 min per day for approximately 3 days. Overall, the habituation period lasts about 6–8 days.

#### Training

Step 1, food restriction. Food restriction includes food deprivation and food limitation. First, the mice were deprived of food for 24 hr to enhance the appeal of the food pellet to the mice while water consumption remained normal. Then, after 24 hr of food deprivation, each cage of mice was given 5 g/mouse of food every morning to meet their daily food requirements until the end of the test.

Step 2, contextual familiarization (*Figure 1B*). At each time after the mice homecage was translocated from the animal facility to the behavioral room, the cage lid was removed to allow the mice to freely explore the cage for about 1–2 min to become accustomed to the open roof. Before training to find the food pellet (step 3), each mouse was allowed to enter the arena chamber from the left side where the mouse was kept for 3 min before being gently driven out from the right side of the chamber to be back to homecage to have a 2 min rest. Then, the mouse would enter the arena chamber from the right side and 3 min later was gently driven out from the left side. Arena familiarization was conducted 1 round per day for 2–3 days.

Step 3, training to find the food pellet (*Figure 1C*). The experimenter placed a small food pellet in the trough in the middle of the floor of chamber. Let one mouse enter the chamber from the left side and stay there until it found and ate the pellet. After that, the mouse was gently driven out of the chamber from the right side to have a 2 min rest in its homecage. Then, the mouse would enter the chamber from the right side and stay there until it found and ate food. The training was repeated 1 round daily until each mouse directly went to the trough and ate the small pellet after entering the chamber (*Figure 1—video 1*).

Step 4, learning to get the food pellet (*Figure 1D*). In this step, the small food pellet in the trough was nearly under the vertical block. As the block was transparent and had holes at lower portions facing the entries, and it was movable thanks to the sliding wheels and the track attached to the roof at the top of the block, the mouse was able to learn to push away the block to get the pellet. The roof attached with track and block was lifted up after the mouse ate the pellet to allow the mouse to go out of the chamber from the side opposite to the entry side. Each mouse was repeatedly trained to get the food pellet alternately from left and right entry until they were able to harvest the pellet directly and easily without competition (*Figure 1—video 2*).

## Test

Test 1, direct win-lose test via analyzing food occupation (*Figure 1F*). Test 1 required several trials, in the first of which the paired contenders entered the chamber from the opposite sides simultaneously. The mice entered the chamber from left or right entry alternately in two consecutive trials. As there was only one small pellet placed under the movable block in each trial, the mouse that obtained the pellet was deemed the winner, while the other one was the loser (*Figure 1—video 3*). To calculate the latency of food harvesting of mice, a starting state was marked when its four legs just entered the chamber arena (*Figure 1—figure supplement 4*). The time spent from starting state to food-getting for a mouse was calculated as latency of food harvesting. One competition test comprised at least four consecutive trials.

Test 2, indirect win-lose test via analyzing mouse's attempt to gain the food (*Figure 1G*). In this test, a slightly larger pellet was placed in the inner bottom of the block, rather than in the trough under the block, so that the mice could see, smell, but not access it. Once the mice entered the chamber, they would push the block and attempt to gain the food. The winner or loser was determined by times of pushing the block in 2 min (*Figure 1—video 4*).

## Tube test

The detailed steps of the tube test were described previously by *Fan et al., 2019*. It consists of the following three steps: adaptation, training, and test. The adaptation phase had been completed at FPCT, referred to as the habituation step of FPCT. In the training phase, the tail of the mouse was gently lifted and placed at one end of the tube, and when the mouse entered the tube, the tail was released and the mouse was allowed to pass through the tube. A plastic rod was used, when the mouse retreated or stagnated for a long time, to gently touch the tail so that the mouse could continue to move until it passed through the tube. The test phase required 4 consecutive days, and the social rank of the mice was ranked for 4 consecutive days. During the experiment, the camera was placed directly in front of the tube, and the whole process of the mouse experiment could be recorded.

## Warm spot test

The experimental apparatus utilized for the WST consisted of a rectangular behavior box with dimensions of 28 cm in length, 20 cm in width, and 40 cm in height. Prior to the commencement of the experiment, the bottom of the box was placed on ice to ensure that its temperature was approximately

0°C. A heating sheet measuring 2 cm by 2 cm was positioned in one of the inner corners of the box, providing sufficient space for only one mouse. The temperature of the heating sheet was maintained at 34°C, and this temperature was monitored using a temperature measuring gun. During the experiment, three mice were introduced into the box simultaneously, allowing them to move freely for a duration of 15 min. The entire experimental procedure was recorded via video, and the time each mouse spent on the heating sheet was subsequently quantified from the recorded footage.

## Statistical analysis

The experimental data were statistically analyzed using Prism 9 or Origin 9.0 software and presented as mean ± SEM. Two-way or one-way ANOVA, unpaired or paired Student's t-test, and Mann-Whitney U test were used to compare the difference between groups. Statistical significance level (p-value) was set at 0.05.

## Additional information

### Funding

| Funder | Grant reference number | Author |
|---|---|---|
| National Natural Science Foundation of China | 82271859 | Rongqing Chen |
| National Natural Science Foundation of China | 31771127 | Rongqing Chen |
| Natural Science Foundation of Guangdong Province | 2023A1515010639 | Rongqing Chen |

The funders had no role in study design, data collection and interpretation, or the decision to submit the work for publication.

### Author contributions

Meiqiu Liu, Conceptualization, Data curation, Formal analysis, Investigation, Visualization, Methodology, Writing – original draft, Writing – review and editing; Yue Chen, Data curation, Formal analysis, Investigation, Methodology; Rongqing Chen, Conceptualization, Data curation, Supervision, Funding acquisition, Visualization, Methodology, Writing – original draft, Project administration, Writing – review and editing

### Author ORCIDs

Meiqiu Liu https://orcid.org/0000-0001-9826-8934
Yue Chen https://orcid.org/0009-0004-6388-8489
Rongqing Chen https://orcid.org/0000-0002-1668-8554

### Ethics

All animal experiments were conducted according to the Regulations for the Administration of Affairs Concerning Experimental Animals (China) and were approved by the Southern Medical University Animal Ethics Committee.

Reviewer #1 (Public review): https://doi.org/10.7554/eLife.103748.3.sa1
Reviewer #2 (Public review): https://doi.org/10.7554/eLife.103748.3.sa2
Reviewer #3 (Public review): https://doi.org/10.7554/eLife.103748.3.sa3
Author response https://doi.org/10.7554/eLife.103748.3.sa4

## Additional files

### Supplementary files

MDAR checklist

## Data availability

All data generated or analysed in this study are included in the manuscript and supporting files; source data files for *Figures 1–4* are provided.

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
