## [Editor Report · eLife Assessment]

This study proposes a **useful** assay to identify relative social ranks in mice incorporating the competitive drive for two basic resources - food and living space. Using this new protocol, the authors provide **solid** evidence of stable ranking among male and female pairs, while reporting more fluctuant hierarchies among triads of males. The evidence is, however, limited by the lack of ethologically based validation, assessment of the influence of competitor recognition, and proof of concept of application to neuroscience. This manuscript may be of interest to those interested in social behavior and related neuroscience.

---

## [Referee Report · Reviewer #1 (Public review)]

Summary

The authors present a new protocol to assess social dominance in pairs and triads of C57BL/6j mice, based on a competition to access a hidden food pellet. Using this new protocol, the authors have been able to identify stable ranking among male and female pairs, while reporting more fluctuant hierarchies among triads of males. Ranking readout identified with this new apparatus was compared to the outcome obtained with the same animals competing in the tube and in the warm spot tests, which have been both commonly used during the last decade to identify social ranks in rodents under laboratory conditions.

Strengths

FPCT allows for an easy and fast identification of a winner and loser in a context of food competition. The apparatus and the protocol are relatively easy and quick to implement in the lab and free from any complex post processing/analysis, which qualifies it for wide distribution, particularly within laboratories that do not have the resources to implement more sophisticated protocols. Hierarchical readout identified through the FPCT correlates with social ranks identified with the tube and the warm spot tests, which have been widely adopted during the last decade and allow for study comparison.

Weaknesses

While the FPCT is validated by the tube and the warm spot test, this paper would have gained strength by providing a more ethologically based validation. Tube and warm spot tests have been shown to provide conflicting results and might not be a sufficient measurement for social ranking (see Varholik et al, Scientific Reports, 2019; Battivelli et al, Biological Psychiatry, 2024). Instead, a general consensus pushing toward more ethological approaches for neuroscience studies is emerging.

Other papers already successfully identified social ranks dyadic food competition, using relatively simple scoring protocol (see, for example, Merlot et al., 2006), within a more naturalistic set-up, allowing the 2 opponents to directly interact while competing for the food. A potential issue with the FPCT, is that the opponents being isolated from each other, the normal inhibition expected to appear in subordinates in presence of a dominant to access food, could be diminished, and usually avoiding subordinates could be more motivated to push for the access to the food pellet.

Comments on revisions:

We thank the authors for the significant improvement of the English in the revised version and for the replacement of some conceptual terms that now seem more relevant and appropriate. We only noticed that the term "society" remains in use, although it might not be appropriate to describe a mouse colony (see previous review).

Conclusive remarks

Although this protocol aims to provide a novel approach to evaluate social ranks in mice, it is not clear how it really brings a significant advance in neuroscience research. The FPCT dynamic is very similar to the one observed in the tube test, where mice compete to navigate forward in a narrow space, constraining the opponent to go backwards. The main difference between the FPCT and the tube test is the presence of food between the opponents. In the tube test, food reward was initially used to increase motivation to cross the tube and push the opponent upon the testing day. This component has been progressively abandoned, precisely because it was not necessary for the mice to compete in the tube.

This paper would really bring a significant contribution to the field by providing a neuronal imaging or manipulation correlate to the behavioral outcome obtained by the application of the FPCT.

---

## [Referee Report · Reviewer #2 (Public review)]

Summary:

In this study, the authors have devised a novel assay to measure relative social rank in mice that is aimed at incorporating multiple aspects of social competition while minimizing direct contact between animals. Forming a hierarchy often involves complex social dynamics related to competitive drives for different fundamental resources, including access to food, water, territory and sexual mates. This makes the study of social dominance and its neural underpinnings hard, warranting the development of new tools and methods that can help understand both social function as well as dysfunction.

Strengths:

This study showcases an assay called the Food Pellet Competition Test, where cagemate mice compete for food, without direct contact, by pushing a block in a tube from opposite directions. This task ran with stranger mice leads to more variable outcomes, suggesting co-housing helps stabilize outcomes. The authors have attempted to quantify motivation to obtain the food independent of other factors by running the assay under two conditions: one where the food is accessible and one where it isn't. This assay results in high outcome consistency across days for females and males paired housed and for male groups of three. Further, the determined social ranks correlate strongly with two common assays: the tube test and the warm spot test.

Weaknesses:

This new assay has limited ethological validity since mice do not compete for food without touching each other with a block in the middle. In addition, the assay may only be valid for a single trial per day, making its utility for recording neural recordings and manipulations limited to a single sample per mouse. The authors claim, as currently stated in results, for the new control experiment in 1H-J is not warranted given that 6/8 mice had majority winning or losing across all strangers.

---

## [Referee Report · Reviewer #3 (Public review)]

Summary:

The laboratory mouse is an ideal animal to study the neural and psychological underpinnings of social dominance behavior because of its economic cost and the animals' readiness to display dominant and subordinate behaviors in simple and testable environments. Here, a new and novel method for measuring dominance and the individual social status of mice is presented using a food competition assay. Historically, food competition assays have been avoided because they occur in an open arena or the home cage, and it can be difficult to assess who gets priority access to the resource and to avoid aggressive interactions such as bite wounding. Now, the authors have designed a narrow rectangular arena separated in half by a sliding floor-to-ceiling obstacle, where the mice placed at opposite sides of the obstacle compete by pushing the obstacle to gain priority access to a food pellet resting on the arena floor under the obstacle. One can also place the food pellet within the obstacle to restrict priority access to the food and measure the time or effort spent pushing the obstacle back and forth. As hypothesized, the outcomes in the food competition test were significantly consistent with those of the more common tube test (space competition) and warm spot competition test. This suggests that these animals have a stereotypic dominance organization that exists across multiple resource domains (i.e., food, space, and temperature). Only male and female C57 mice in same-sex pairs or triads were tested.

Strengths:

The design of the apparatus and the inclusion of females are significant strengths within the study.

Weaknesses:

There are at least two major weaknesses of the study: the test with unfamiliar non-cagemates and not providing the mice time to recognize who they are competing with.

The authors conclude in the first section of the results that they "did not detect significant difference in winning/losing results between unfamiliar non-cagemate male mice." Given the data and analysis provided, I believe this statement is false. My understanding is that the authors would like to show that the establishment of social relationships (i.e., familiarity) is necessary for FPCT to distinguish social ranks of mice. There are many ways to test this. The simplest would be to randomly pair unfamiliar non-cagemates that are housed in isolation with one another and see if they perform at chance, individually. The more involved empirical way would be to measure the ranks of mice in a social group, then test them with unfamiliar non-cagemate mice to see if they maintain their outcomes regardless of social familiarity, or return to chance outcomes when paired with non-cagemates. Figure 1I clearly shows that they did not perform at chance. Since the outcome is win or lose, then the probability of getting all of one outcome 4 times in a row would be 1 in 16. The data shows that this occured twice, so 2 mice of 8 had the same outcome 4 times in a row (i.e., Mouse B3 and A1). So, they did not perform at chance. I am not even sure if there are enough animals here to test this question. One may need to consult a mathematician. Moreover, the original tube-test study by Lindzey et al. 1961 (https://www.nature.com/articles/191474a0) used unfamiliar non-cagemate male mice, and showed that 100% of the A/alb strain won more than half of their oppositions against C3H and DBA/8 mice. Thus, A/alb mice were more "dominant" mice relative to C3H or DBA/8. Taking into consideration the results, is mouse A1 naturally dominant? So maybe it doesn't matter what mouse you pair with it, it will always win? If this is true, is "individual identification of the partner" actually necessary to get this outcome? All they have to do is push to get the food reward, does it matter who is on the other side? If one wants to measure social dominance relationships, then it should matter who is on the other side. If one would like to measure attributes of dominant behavior (e.g., pushing), then one may do so and not insinuate a social link. Studying dominance relationships (i.e., social ranking) of animals is an extremely difficult task. We must ensure that we are not assigning something about a relationship that does not exist. Please read "Dominance: The baby and the bathwater" but Irwin Bernstein, https://annas-archive.org/scidb/10.1017/s0140525x00009614/

Unlike the tube test and warm spot test, the food competition test presented here provides no opportunity for the animals to identify their opponent. That is, they cannot sniff their opponent's fur or anogenital region, which would allow them an opportunity to identify them individually. Thus, as the authors state, the test only measures a psychological motivation to get a food reward. Notably, the outcome in the direct and indirect testing of food competition is in agreement, leaving many to wonder whether they are measuring the social relationship or the effort an individual puts forth in attaining a food reward regardless of the social opponent. Specifically, in the direct test, an individual can retrieve the food reward by pushing the obstacle out of the way first. In the indirect test, the animals cannot retrieve the reward and can only push the obstacle back and forth, which contains the reward inside. In Figure 2F, you can see that winners spent more time pushing the block in the indirect test--albeit not significantly. Thus, whether the test measures a social relationship or just the likelihood to gain priority access to food is unclear. To rectify this issue, the authors could provide an opportunity for the animals to interact before lowering the obstacle and raising(?) a food reward. They may also create a very long one-sided apparatus to measure the amount of effort an individual mouse puts forth in the indirect test with only one individual-or any situation with just one mouse where the moving obstacle is not pushed back, and the animal can just keep pushing until they stop. This would require another experiment. It also may not tell us much more since it remains unclear whether inbred mice can individually identify one another (see https://doi.org/10.1098/rspb.2000.1057 for more details).

---

## [Author Response]

The following is the authors’ response to the original reviews.

**Reviewer #1 (Public review):**
Summary:The authors present a new protocol to assess social dominance in pairs and triads of C57BL/6j mice, based on a competition to access a hidden food pellet. Using this new protocol, the authors have been able to identify stable ranking among male and female pairs, while reporting more fluctuant hierarchies among triads of males. Ranking readouts identified with this new apparatus were compared to the outcomes obtained with the same animals competing in the tube and in the warm spot tests, which have been both commonly used during the last decade to identify social ranks in rodents under laboratory conditions.Strengths:FPCT allows for easy and fast identification of a winner and a loser in the context of food competition. The apparatus and the protocol are relatively easy and quick to implement in the lab and free from any complex post-processing/analysis, which qualifies it for wide distribution, particularly within laboratories that do not have the resources to implement more sophisticated protocols. Hierarchical readouts identified through the FPCT correlate with social ranks identified with the tube and the warm spot tests, which have been widely adopted during the last decade and allow for study comparison.Weaknesses:While the FPCT is validated by the tube and the warm spot test, this paper would have gained strength by providing a more ethologically based validation. Tube and warm spot tests have been shown to provide conflicting results and might not been a sufficient measurement for social ranking (see Varholik et al, Scientific reports, 2019; Battivelli et al, Biological psychiatry, 2024). Instead, a general consensus pushing toward more ethological approaches for neuroscience studies is emerging.

We appreciate all the reviewers for recognizing the strength of the FPCT setup and the data. We also appreciate the reviewers for pointing out weakness and giving us valuable suggestions that help us to improve the quality of our manuscript through revision.

In this manuscript, we found the ranking results of the FPCT were largely consistent with the tube and the warm spot tests. Such a finding was unexpected by us as we considered that different competitive targets of different paradigms should provide the mice with distinct appeals and enable them to exert their specific advantages. However, the consistency between the FPCT and tube test was observed in the pairs of female mice, pairs of male mice and triads of male mice. The consistency between the FPCT, tube test and warm spot test was observed in pairs of male mice and triads of male mice. Thus, we concluded that there is a social rank-order stability of mice.

We acknowledge that it’d better if this conclusion could be validated by more ethological approaches like urine-marking analysis and water competition test. Whereas, we did not rule out inconsistency of ranking results between two or more paradigms. Actually, there were inconsistent cases in our experiments. The inconsistency of ranking results between paradigms, even between FPCT and tube test, could be amplified if the tests were operated with other details of experimental protocols and conditions. This is in that too many factors and aspects can affect the readouts, such as formation of colony, tasks, test protocols, habituation and training. Using tube test itself, both stable 1,2 and unstable 3 ranking results have been reported.

Other papers already successfully identified social ranks dyadic food competition, using relatively simple scoring protocol (see for example Merlot et al., 2006), within a more naturalistic set-up, allowing the 2 opponents to directly interact while competing for the food. A potential issue with the FPCT, is that the opponents being isolated from each other, the normal inhibition expected to appear in subordinates in the presence of a dominant to access food, could be diminished, and usually avoiding subordinates could be more motivated to push for the access to the food pellet.

The hierarchical structure of mice colony could be established on the basis of physical aspects—such as muscular strength, vigorousness of fighting—and psychological aspects— such as boldness, focused motivation, active self-awareness of status. In the contexts of currently available food contest paradigms where the mice compete with bodily interaction, the physical and psychological aspects are intermingled in the interpretation of the mice’s winning/losing. In the FPCT, the opponents are isolated from each other so that the importance of direct bodily interaction in a competition is minimized, facilitating the exposure of psychological factors contributing to the establishment and/or expression of social status of the mice. In this study, the overall stable ranking results across the FPCT, tube test and warm spot test indicate that the status sense of animals is part of a comprehensive identify of self-recognition of individuals in an established mice social colony.

There are issues with use of the English language throughout the text. Some sentences are difficult to understand and should be clarified and/or synthesized.

We thank the reviewer for pointing out language issues. We have carefully corrected the grammar errors.

Open question:Is food restriction mandatory? Palatable food pellet is not sufficient to trigger competition? Food restriction has numerous behavioral and physiological consequences that would be better to prevent to be able to clearly interpret behavioral outcomes in FPCT (see for example Tucci et al., 2006).

We thank the reviewer for raising this question. In the preliminary experiments, we noticed that food restriction was mandatory and palatable food pellet was not sufficient to trigger competition. In order to limit the potential influence of food restriction on competitive behavior, the mice underwent only a 24-hour food deprivation period at the beginning of training, followed by mild restriction of food supply to meet basic energy requirement.

Conclusive remarks:Although this protocol attempts to provide a novel approach to evaluate social ranks in mice, it is not clear how it really brings a significant advance in neuroscience research. The FPCT dynamic is very similar to the one observed in the tube test, where mice compete to navigate forward in a narrow space, constraining the opponent to go backward. The main difference between the FPCT and the tube test is the presence of food between the opponents. In the tube test, a food reward was initially used to increase motivation to cross the tube and push the opponent upon the testing day. This component has been progressively abandoned, precisely because it was not necessary for the mice to compete in the tube.This paper would really bring a significant contribution to the field by providing a neuronal imaging or manipulation correlate to the behavioral outcome obtained by the application of the FPCT.

Thank the reviewer for this comment on the significance of the FPCT paradigm. In this manuscript, we think it is interesting to report that the ranking results were consistent across the FPCT, tube test and warm spot test. This finding indicates that the status sense of animals might be a part of a comprehensive identify of self-recognition of individuals in an established social colony.

Moreover, we are conducting researches on biological consequences and mechanisms of social competition. Hopefully, the results of the on-going project will be published in the near future.

**Reviewer #2 (Public review):**
Summary:In this study, the authors have devised a novel assay to measure relative social rank in mice that is aimed at incorporating multiple aspects of social competition while minimizing direct contact between animals. Forming a hierarchy often involves complex social dynamics related to competitive drives for different fundamental resources including access to food, water, territory, and sexual mates. This makes the study of social dominance and its neural underpinnings hard, warranting the development of new tools and methods that can help understand both social functions as well as dysfunction.Strengths:This study showcases an assay called the Food Pellet Competition Test where cagemate mice compete for food, without direct contact, by pushing a block in a tube from opposite directions. The authors have attempted to quantify motivation to obtain the food independent of other factors such as age, weight, sex, etc. by running the assay under two conditions: one where the food is accessible and one where it isn't. This assay results in an impressive outcome consistency across days for females and males paired housed and for male groups of three. Further, the determined social ranks correlate strongly with two common assays: the tube test and the warm spot test.Weaknesses:This new assay has limited ethological validity since mice do not compete for food without touching each other with a block in the middle. In addition, the assay may only be valid for a single trial per day making its utility for recording neural recordings and manipulations limited to a single sample per mouse. Although the authors attempt to measure motivation as a factor driving who wins the social competition, the data is limited. This novel assay requires training across days with some mice reaching criteria before others. From the data reported, it is unclear what effects training can have on the outcome of social competition. Beyond the data shown, the language used throughout the manuscript and the rationale for the design of this novel assay is difficult to understand.

We appreciate the reviewers for the valuable comments on the strength and weakness of our manuscript.

The design mentality of the FPCT was to (1) provide researchers with a choice of new food competition paradigm and (2) expose psychological factors influencing the establishment and/or expression social status in mice by avoiding direct physical competition between contenders (see revised Abstract and the last paragraph in the Introduction).

As a result, the consistent ranking across the FPCT, tube test and warm spot test might indicate that the status sense of animals is part of a comprehensive identify of self-recognition of individuals in an established social colony.

We suggest to perform the FPCT test one trial per day per mouse as the mice might lose interest in the food pellet if it is tested frequently in a day, but it is practical to perform the FPCT assay for several days.

Regarding the training, we suggest 4-5 days for training as we did. In this revision, we add training data which show the progressing latency of food-getting of mice (Figure 1). At the last day of training, the mice would go directly to push the block and eat the food after they entered the arena.

We thank the reviewer for pointing out language issues. We have carefully corrected the errors.

**Reviewer #3 (Public review):**
Summary:The laboratory mouse is an ideal animal to study the neural and psychological underpinnings of social dominance behavior because of its economic cost and the animals' readiness to display dominant and subordinate behaviors in simple and testable environments. Here, a new and novel method for measuring dominance and the individual social status of mice is presented using a food competition assay. Historically, food competition assays have been avoided because they occur in an open arena or the home cage, and it can be difficult to assess who gets priority access to the resource and to avoid aggressive interactions such as bite wounding. Now, the authors have designed a narrow rectangular arena separated in half by a sliding floor-to-ceiling obstacle, where the mice placed at opposite sides of the obstacle compete by pushing the obstacle to gain priority access to a food pellet resting on the arena floor under the obstacle. One can also place the food pellet within the obstacle to restrict priority access to the food and measure the time or effort spent pushing the obstacle back and forth. As hypothesized, the outcomes in the food competition test were significantly consistent with those of the more common tube test (space competition) and warm spot competition test. This suggests that these animals have a stereotypic dominance organization that exists across multiple resource domains (i.e., food, space, and temperature). Only male and female C57 mice in same-sex pairs or triads were tested.Strengths:The design of the apparatus and the inclusion of females are significant strengths within the study.Weaknesses:There are at least two major weaknesses of the study: neglecting the value of test inconsistency and not providing the mice time to recognize who they are competing with.Several studies have demonstrated that although inbred mice in laboratory housing share similar genetics and environment, they can form diverse types of hierarchical organizations (e.g., loose, stable, despotic, linear, etc.) and there are multiple resource domains in the home cage that mice compete over (e.g., space, food, water, temperature, etc.). The advantage of using multiple dominance assays is to understand the nuances of hierarchical organizations better. For example, some groups may have clear dominant and subordinate individuals when competing for food, but the individuals may "change or switch" social status when competing for space. Indeed, social relationships are dynamic, not static. Here, the authors have provided another test to measure another dimension of dominance: food competition. Rather than highlight this advantage, the authors highlight that the test is in agreement with the standard tube test and warm spot test and that C57 mice have stereotypic dominance across multiple domains. While some may find this great, it will leave many to continue using the tube test only (which measures the dimension of space competition) and avoid measuring food competition. If the reader looks at Figures 6E, F, and G they will see examples of inconsistency across the food competition test, tube test, and warm spot test in triads of mice. These groups are quite interesting and demonstrate the diversity of social dynamics in groups of inbred mice in highly standardized environmental conditions. Scientists interested in dominance should study groups that are consistent and inconsistent across multiple dimensions of dominance (e.g., space, food, mates, etc.).Unlike the tube test and warm spot test, the food competition test presented here provides no opportunity for the animals to identify their opponent. That is, they cannot sniff their opponent's fur or anogenital region, which would allow them an opportunity to identify them individually. Thus, as the authors state, the test only measures psychological motivation to get a food reward. Notably, the outcome in the direct and indirect testing of food competition is in agreement, leaving many to wonder whether they are measuring the social relationship or the effort an individual puts forth in attaining a food reward regardless of the social opponent. Specifically, in the direct test, an individual can retrieve the food reward by pushing the obstacle out of the way first. In the indirect test, the animals cannot retrieve the reward and can only push the obstacle back and forth, which contains the reward inside. In Figure 4E, you can see that winners spent more time pushing the block in the indirect test. Thus, whether the test measures a social relationship or just the likelihood of gaining priority access to food is unclear. To rectify this issue, the authors could provide an opportunity for the animals to interact before lowering the obstacle and raising(?) a food reward. They may also create a very long one-sided apparatus to measure the amount of effort an individual mouse puts forth in the indirect test with only one individual - or any situation with just one mouse where the moving obstacle is not pushed back, and the animal can just keep pushing until they stop. This would require another experiment. It also may not tell us much more since it remains unclear whether inbred mice can individually identify one another(see https://doi.org/10.1098/rspb.2000.1057 for more details).A minor issue is that the write-up of the history of food competition assays and female dominance research is inaccurate. Food competition assays have a long history since at least the 1950s and many people study female dominance now.Food competition: https://doi.org/10.1080/00223980.1950.9712776, https://psycnet.apa.org/fullte xt/1953-03267- 001.pdf,https://doi.org/10.1016/j.bbi.2003.11.007,https://doi.org/10.1038/s41586-02204507-5Female dominance: history https://doi.org/10.1016/j.cub.2023.03.020, https://doi.org/10.1016/S0 031-9384(01)00494-2, https://doi.org/10.1037/0735-7036.99.4.411

We thank the reviewers very much for so many helpful comments and suggestions.

In this manuscript, we want to address the overall and averagely consistency of ranking results between FPCT, tube test and warm spot test as an unexpected finding. We agree that the inconsistency of social ranking occurred between trials and between paradigms should not be ignored. In the revision, we added description and discussion of inconsistent part of the different test paradigms (paragraph 2 in the section 3 of the Result, last 2 sentences of paragraph 4 in the Discussion)

Although the two opponents were separated each other, they were able to see and sniff each other because the block is transparency, there are holes in the lower portion of the block, and there is the gap between the block and chamber (Supplementary figures 1 and 2). In the female but not male groups, the presence of a cagemate opponent during the test 1 could significantly disturb the female mice and increase the its latency to get the food, comparing with last day of training when there was no opponent (Figure 3A). This indicates that one mouse, at least female mouse, could identify the existence of the opponent in the opposite side of the chamber. To further see whether social relation was influential to readouts of the FPCT, we performed additional experiments using two groups of non-cagemate mice to perform the competition. We did not detect obviously different ranks between the two groups (Figure 1H-1J), suggesting that establishment of social colony is necessary for FPCT to distinguish social ranks of mice.

Thank the reviewer for reminding us to recognize the history of food competition assays. We have added the citations and discussions of related literatures, both for male (paragraph 2 in the Introduction; paragraph 3 in the Discussion) and female (paragraph 1 of section 3 in the Results; paragraph 4 in the Discussion) mice.

**Reviewer #1 (Recommendations for the authors):**
There are issues with use of the English language throughout the text. Some sentences are difficult to understand and should be clarified and/or synthesized.

We appreciate the reviewer for constructive comments and helpful corrections.

“Despite that 6 in 9 groups of mice display some extent of flipped ranking (Figures 6B-6G) and only 3 in 9 groups displayed continuously unaltered ranking (Figure 6H) during a total of 9 trials consisting of 3 trials of FPCT, 3 trials of tube test and 1 trial of WST, an obvious stable linear intragroup hierarchy was observed throughout all the trials and tasks"

The above sentence has been re-written as: The ranking result showed that 6 in 9 groups of mice displayed some extent of flipped ranking (Figures 4B-4G), and only 3 in 9 groups displayed continuously unaltered ranking (Figure 4H). Averagely, in the totally 27 trials consisting of 12 trials of FPCT, 12 trials of tube test and 3 trials of WST, an obvious stable linear intragroup hierarchy was observed across all the trials and tasks (paragraph 1 of section 4 in the Results).

"it is hard to attribute winning a competition in a shared space to stronger motivation rather than muscular superiority".

The above sentence has been deleted and re-written in paragraph 1 of section 4 in the Results and paragraph 3 in the Discussion.

"Unexpectedly, in most of the trials the mice preserved the winner or loser identity acquired in FPCT into tube test and WST (Figures 5L-5O)".Why this is unexpected? Instead, it looks like this result is expected (tube test has been successfully applied to identify ranks in females, see Leclair et al, eLife, 2021).

We thank the reviewer for raising this point. FPCT is different from tube test and warm spot test at least in two aspects: competition for food vs space; presence vs absence of direct bodily interaction during competition. Some mice might be active in food competition, but not in space competition, while others might be on the contrary. Some mice might be good at physical contest, while others might be good at play tricks. Therefore, these factors made us expect task-specific outcomes of ranking results.

Vocabulary issues:"Stereotypic", to talk about rank stability in a different context does not look appropriate. In behavioral neuroscience, stereotypy is more excepted to intend abnormal repetitive behaviors. The stability that the authors seem to indicate with the word "stereotype" refers rather to the concept of "consistency" or "stability".

We thank the reviewer for this detailed explanation. We have chosen to use "stability" to describe the data.

"Society", to talk about groups or colonies of animals sounds a bit odd. Society evokes more abstract concepts more likely to fit with human organization. I suggest the use of "group" or "colony"."Hide" to qualify the block preventing access to the food pellet. It is said that the block is transparent. We suggest the use of "inaccessible" instead of hidden.We strongly encourage the authors to further edit the entire script to improve language.

Thank the reviewer for kind correction. We have corrected the above vocabulary misuse.

Technical issues / typos:Figure 1. The picture does not seem optimal to visualize the apparatus.Missing unit legend in Figure 4E.Supplementary videos 2 and 4 are missing.

We have added a frontal view of the apparatus in the figure (Supplementary Figure 1), added a unit to the Figure 2F (previous Figure 4E), and we will make sure to upload the missing videos.

**Reviewer #2 (Recommendations for the authors):**
While the assay shows promise as a tool for studying social dominance, the study suffers from some limitations such as lack of ethological relevance. In addition, there is a lack of rationale and methodological clarity in the manuscript that can impact the ability of other scientists to be able to perform this novel assay.(1) Related to lack of scientific rigor:a. In the first paragraph of the introduction, the authors mention that "disability in social recognition and unsatisfied social status are associated with brain diseases such as autism, depression and schizophrenia". Both papers that they cited refer to mouse models, not humans (which is the species that is attributed these diagnoses clinically). In addition, neither citation discusses schizophrenia. While social dysfunctions can indeed be related to these diseases, to my knowledge this is not caused by a change in "social status" and there is no human data with patient populations and social status. Therefore, this sentence is inaccurate and there is no research that demonstrates that.

We thank the reviewer for raising this point. To express the opinion and cite literatures more accurately, we improved the sentence in the 1st paragraph of Introduction as follows: “Impaired awareness of social competition has been documented in individuals with autism spectrum disorder (ASD)4,5, and reduced social interaction has been characterized in corresponding animal models6. Similarly, maladaptive responses to social status loss has been associated with patient depressive disorders7,8 and animal models of depression1,9”. The reviewer is right that no patient disease is causally related with social status, and only depression has been proposedly associated with change of social status7,8.

b. In the second paragraph of the introduction, the authors mention a scarcity of research papers with designs for food competition-based social hierarchy assays for mice. At least two such papers have been published in the past few years (DOIs https://doi.org/10.1038/s41586- 021-04000-5 and https://doi.org/10.1038/s41586-022-04507-5). The authors should acknowledge the existence of these and other assays and discuss how their work would be related. In the same paragraph, they also mention that existing assays suffer from "hierarchy instability" and "complex calculations" without showing any citations or details for these claims.

We thank the reviewer for raising this point. We acknowledged that there are some available food competitions to measure social hierarchy for mice. But relative to space competition, food competition tests have not been used so commonly and widely. No food competition paradigm has been accepted as generally as some space competition paradigms like tube test and warm spot test. To improve the language and scientific expression, we revised the sentences as follows: “Relative to space competition, food competition tests for mice have been designated and applied less commonly in animal studies despite its long history 28-30. Several issues could be thought to be the underlying limitations for the application of food competition paradigms. First, there are methodological issues in some of these approaches, such as long video recording duration and difficulty in analyzing animal’s behaviors during competitive physical interaction in videos, hindering their application by laboratories that cannot afford sophisticated equipment and analysis”. Corresponding citations have been updated (see paragraph 3 in the Introduction).

c. The authors say that their study is the first to demonstrate that female mice follow social ranks. This is not the first study to do so and the authors should acknowledge existing publications that have done the same (eg DOI https://doi.org/10.7554/eLife.71401).

We have followed the reviewer’s suggestion to increase citations regarding social ranking of female mice tested by competition paradigms, especially food competition paradigms (see paragraph 1 of section 3 in the Results; paragraph 4 in the Discussion).

(2) Related to problems with interpretation of data:a. The authors showed the assay works for females and males in pairwise housing, but two mice don't make a hierarchy, as hierarchies require a minimum of three individuals. Therefore, whether the assay works for females caged in three is an important question that is unaddressed in this study and is a caveat. extended the competition assay to male mice that are housed in cages of three. It would be important to show whether the assay generalizes well for female mice with this three-animal housing as well as discuss the effect of using even bigger groups of mice on the results of the assay.

We thank the reviewer for raising questions related to the interpretation of data and giving us the insightful the suggestions. We agree that it is interesting and important to probe if FPCT works for a group of three female mice. Although social rankings of pairs of male and female mice were not significantly different (new Figure 2D-2F and 3F-3H), that of triads of male and female mice could be different. We have tested trads of male mice and found that the mice displayed an overall linear hierarchical ranking. We would like to use FPCT to investigate the rankings of trads of female mice and even bigger group of mice in the future. In the present manuscript we’d like to address the feasible application of the FPCT in smaller groups. In the Discussion, we add contents commenting group size effect on social competition tests (see paragraph 4 in the Discussion).

b. The authors claim that "test 2" of their assay helps assert the motivation of mice for social competition as in Figure 4E. This could simply be a readout of how strong the mice are (muscle mass). To claim that this is indeed related to motivation during the FPCT assay, the authors should show the correlation of this readout with the latency to push the block during the social competition task.

We appreciate the reviewer for raising this question. The dimensions establishing the social structures include physical and psychological factors. In the FPCT paradigm, the two contenders are separated so that physical factors are minimized in this context and psychological factors should play more important role in competition in comparison with previous reported food competition paradigms. Therefore, in the revised manuscript we consider to attribute the ranking results mainly to psychological factors, rather than only motivation which is just one of the numerous psychological factors (paragraph 3 of Discussion). Moreover, in the Discussion we point out that we could not exclude physical factors still participate in the determination of competitive outcomes since some of mice pairs pushed the block simultaneously (paragraph 3 of Discussion).

c.The authors mention that they are interested to understand which factors lead to the outcome of the competition such as age, sex, physical strength, training level, and intensity of psychological motivation. However, in all their runs of the assay, they always matched these variables between the competitors. They should clarify that they were instead controlling for these variables. Another thing to note here is that while they controlled the body mass of the animals, that isn't the same as physical strength, as a lighter mouse can have more muscle mass than a heavier mouse. They should either specify this limitation or quantify the additional metric of "muscle mass" which is a much better proxy for physical strength. Thus, the claim that the outcome of the competition is solely affected by motivation is not convincing since they didn't rule out the others such as quantifying the rate of learning during training and strength.

We thank the reviewer for addressing this question. As our response to the question in (c), we acknowledge that it is not accurate to ascribe the outcomes of FPCT to psychological motivation. In the revised manuscript, the dimensions of contributing factors to the outcomes of FPCT have been simplified to physical and psychological factors. We consider that the psychological factor could be the main driver of mice participating in FPCT (see paragraph 3 of Discussion).

d. In the discussion, the authors mention that their task only requires a single day of food deprivation (the day before the first trial) while other assays suffer from a continued food deprivation protocol. However, the authors also use 10g per cage as the amount of food instead of giving them ad libitum access. Limited food is a food deprivation method. Thus, this is an inaccurate claim.

We thank the reviewer for raising this point. We have clarified the requirement of food restriction for FPCT in the revision. The mice were deprived of food for 24 hours while water consumption remained normally to enhance the appeal of the food pellet to the mice. Then, after 24 hours of food deprivation, each cage of mice was given 10 g of food every morning to meet their daily food requirements until the end of the test (see FPCT procedure section in Methods and materials).

e.In the second section of the results, the authors run their assay with female mice that are housed in cages of two. This section suffers from the same limitations as the first and can be improved by showing the training data, correlations of competition outcome with "motivation" and ruling out the other factors that could contribute to the outcome. Further, the authors saying that their FPCT assay is enough to show that female mice follow a social hierarchy by itself is a weak claim. They should instead include their cross-validation with the others to strengthen it.

We appreciate the reviewer for raising this question. We have taken the reviewer’s suggestion to show the training data (Figures 1E, 2A and 3A). As the factors contributing to the outcomes of FPCT are diverse, we’d like not to control and determine the exact factor in the current manuscript. We agree with the reviewer that cross-validation with different paradigms is suggested for the studies to rank social hierarchy as the ranking results could be variable with tasks, procedures and operations.

f. In the last paragraph of the introduction, the authors mention how their assay involves "peaceful competition" since the mice are not in direct contact and hence cannot exhibit aggression. The authors do not address the limitation that a lack of physical contact actually makes the assay less ethological. Further, since the mice are housed in groups of two and three, it is not guaranteed that the mice will not be aggressive during their time in the home cage, which could affect their behavior during the competition assay. Whether the assay causes more aggression in the cage due to the lack of physical contact during the competition is not addressed in this study.

We thank the reviewer for raising this point. Diverse factors affect the outcomes of a food competition test, some of which belong to psychological factors and others belong to physical factors. We agree that a lack of physical contact makes the assay less naturally ethological. However, when the social statuses have been established during habituation housing a group of mice for enough time, the win/lose outcomes in the FPCT could be a readout of the expression of social statuses since the mice cannot exhibit aggression in the test. We have revised the Introduction and Discussion (paragraph 3 of Discussion). Thank you.

(3) Related to lack of methodological rigor and rationale clarity:

a. In the first section of the results, the authors run their assay with male mice that are housed in cages of two. While the data that they display is promising, we do not see how mice change behavior across days of training and how that relates to the outcome of the competition. It would be valuable to also show the training data for the mice, answering questions related to competency and any inter-animal variabilities prior to rank assessment. Plotting the training data across all days would be helpful for the other parts of the results as well. This is especially important because the methods mention that mice are trained until they get to the criterium, so this means that different individuals get different amounts of training.

We appreciate the reviewer for addressing the importance of showing training data. We have taken the reviewer’s suggestion and shown the training data (Figures 1E, 2A and 3A).

b. It is unclear why the assay was run only once per mouse pair per day since most protocols for the tube test involve multiple repetitions each day while alternating the side from which the mice enter. The authors should address whether a single trial per day is enough to show consistent results and that it wouldn't vary with more.

We suggest to run the FPCT once or twice per mouse per day under conditions of mild food restriction, training and test procedures in this manuscript. Frequent tests might make the mice’s interest in the food pellet gradually diminished because the food supply was not fully deprived. According to our data, the outcomes of FPCT in 4 consecutive days were overall stable.

c. In the results the authors say that they "raised 3 male mice" which may be incorrect because they report in the methods buying the mice buy mice and they housed all their mice for only three days before running the assay which might be too little for the hierarchy to stabilize. The authors should comment on what was the range of the cohabitation across different cages and whether it had an impact on the results.

According to our experiments, housing the mice for 3 days is enough to establish a mice social colony with relative stable status structure. Prolonged housing may produce either similar, stabler or more dynamic social colony.

d. There are also some formatting and/or convention issues in the results. The first figure callout in the results is for Figure 4 instead of Figure 1 (which is the standard). This is because the authors do not explain how the mice are trained for the task in the results section and show limited data about the training of the task. Not showing comprehensive training data would make replication of this study very difficult.

We appreciate the reviewer for raising this question. We have re-arranged the figures. The new arrangement of figures started with schematic drawing of FPCT procedure and training data (Figure 1).

e. The authors don't report the exact p-values in the figures

We reported the difference level in the figures in the revised manuscript. Thank you.

4. The writing of the manuscript suffers from a lack of clarity in most sections of the manuscript.Here are several examples that are critical:a. In the title and abstract, it isn't clear what the authors mean by "stereotype". It could be a behavior during the competition, or that the social ranks across assays are correlated or that the rank for the new assay is consistent across days.b. There are several instances where the authors anthropomorphize mice using human features such as "urbanization" and "society" which are not established factors affecting mouse hierarchy. This further extends to anthropomorphizing mice in ways that are not standard such as an animal being "timid" or "bold" which would be hard to measure in mice, if not impossible.c. Across the social dominance literature, relative social rank is described using more general "dominant" and "subordinate" titles instead of "superior" and "inferior" that are sometimes used in the manuscript. The authors should follow the standard language so that readers understand.d. In the third paragraph of the introduction, the authors say "Thus, it is more likely expected that different paradigms to weigh the social competency and status may lead to diverse readouts, given that competitive factors are included in competition paradigms." This sentence suffers from multiple syntax errors thereby reducing claritye. There are several typos in the manuscript such as using "dominate" instead of "dominant", "grades" instead of "outcomes" and "forth" instead of "fourth", to give a few examples.

We thank the reviewer for careful reading of the manuscript and very helpful comments. We have taken the above suggestions and improved the writing of the manuscript. For examples, "stereotype" was replaced by “stability”, mice "society" was expressed by "colony", the sentence “Thus, it is more.... in competition paradigms” has been deleted.

**Reviewer #3 (Recommendations for the authors):**
(1) The justification for the design of this new test paradigm is unclear. In the abstract, you state that the field needs a reliable, valid, and easily executable test. Your test provides this, as you state, but how is it better than the tube test? Does the tube test suffer from taskspecific win-or-lose outcomes? Can you provide evidence for this? The nature methods protocol for the tube test (https://doi.org/10.1038/s41596-018-0116-4) "strongly suggest using more than two dominance measures, for example, by also carrying out the warm spot test, or territory urine marking or ultrasonic courtship vocalization assays." This would suggest that results from the tube test can be task-specific, but I am not convinced that you have demonstrated that results from your food competition test are not task-specific. Indeed, by your title, one must run multiple tests.This same problem is apparent in the introduction. In the second paragraph, there is a discussion of the tube test, warm spot test, and food competition tests. What is the problem with these tests?I believe that social dominance relationships are complex and dynamic social relationships indicating who has priority access to a resource between multiple animals that live together. In these living situations, several resources can often be capitalized competed over-for example, space, food, mates, temperature, etc. Currently, we have tests to measure space via the tube test or urine marking, mates via ultrasonic vocalization, temperature via warm spot test, and food via food competition assays. The tube test, urine marking assay, and ultrasonic vocalization test have been demonstrated to be reliable, valid, and easily executable. However, the food competition assays are often difficult to execute because it is difficult to interpret the dominant behaviors and aggressive behaviors like bite wounding can occur during the test. Here, you present a new food competition assay to address these issues and show that it can be used in conjunction with other assays to measure social dominance across multiple resources easily. In doing so, you revealed that many same-sex groups of C57 mice have a stereotypic pattern of dominance behavior when competing across multiple types of resources: space, temperature, and food.I ask that you please rebut if you disagree with me, and adjust your abstract, introduction, and discussion accordingly.

We thank the reviewer for all the constructive comments. We have adjusted the Abstract, Introduction and Discussion of the manuscript.

We recognize and appreciate the valuable tube test, warm spot test and many other competition tests, including food competitions. Tube test and warm spot test are space competition tasks. Relative to space competition, food competition tests for mice have been designated and applied less commonly in animal studies. Several issues (such as methodological issue, aggressive behaviors occurring in competition, and prolonged food deprivation) could be thought to be the underlying limitations of the application of food competition paradigms (paragraph 3 in the Introduction). Therefore, we clarify that the justification for the design of FPCT was “to have a new choice of food competition paradigm for mice, and to facilitate the exposure of psychological aspects contributing to the winning/losing outcomes in competitions” (last paragraph in the Introduction).

FPCT is different from tube test and warm spot test at least in two ways. FPCT is food completion task where the mice need no physical contact during competition, while tube test and WST are space competition tasks where the mice need direct physical contact during competition. Therefore, we expected inconsistent evaluation results of competitiveness and rankings if we compared FPCT with typically available competition paradigms—tube test and WST (last paragraph in the Introduction).

(2) The design of the test needs to be described before the results. You can either move the methods section before the results or add a paragraph in the introduction to better describe the test. Here, you can also reference Figures 1 through 3 so that the figures are presented in the order of which they are mentioned in the paper. (It is very confusing that the first reference to a figure is Figure 4, when it should be Figure 1).

We appreciate the reviewer for raising this point and giving us suggestions. We have added a new section (section 1) in the Results. In the revised manuscript, the figures in the Results start with Figure 1 which shows schematic drawing of FPCT procedure, training data and some test results (Figure 1).

(3) The sentence describing Figure 4H. You argue that this shows that the mice are well and equally trained. It also shows that they have the same motivation or preference for the food.

We appreciate the reviewer for this helpful comment. Data in previous Figures 4H and 5I have been presented as new Figures 2A and 3A, respectively, of revised manuscript. These retrospect analysis of training data displayed similar training level of food-getting and craving state for food (Sections 2 and 3 in the Results).

(4) "Social ranking of multiple cagemate mice using FPCT, tube test and WST"Here, you claim that "comparison of inter-task consistency revealed that the ranks evaluated by FPCT, tube test and WST did not differ from each other...Figure 6K." Okay, however, it is important to discuss the three cases when there wasn't consistency between the tests! Figure 6E-G.

We appreciate the reviewer for raising this point. In the revised manuscript, we add description and discussion of inconsistent part of the different test paradigms (paragraph 2 in the section 3 of the Result, last 2 sentences of paragraph 4 in the Discussion)

(5) Replace all instances of "gender" with "sex". Animals do not have a gender.(6) Adjust the strain of the mice to C57BL/6JNifdc.

We have replaced "gender" with "sex" and “C57BL/6J” with “C57BL/6JNifdc”. Thank you for your careful correction.

(7) What is the justification for running the warm spot test for one day and the other tests for four days?

From the consecutive FPCT and tube test, we already knew that the ranking results were overall stable. This stability was still observed in the day of warm spot test. A bad point for frequent warm spot test is that mice get much stress due to exposure in ice-cold environment. Therefore, we terminated the competition test after only one trial of warm spot test.

(8) GrammarThe second sentence of the abstract: ...recognized as a valuable...Results, sentence after "...was observed (Figure 4G)." it should be "Fourth"

We have corrected these and other grammar errors. We appreciate the reviewers for very careful review and all helpful comments.